# γδ T Cells: The Ideal Tool for Cancer Immunotherapy

**DOI:** 10.3390/cells9051305

**Published:** 2020-05-24

**Authors:** Mahboubeh Yazdanifar, Giulia Barbarito, Alice Bertaina, Irma Airoldi

**Affiliations:** 1Stem Cell Transplantation and Regenerative Medicine, Department of Pediatrics, Stanford University School of Medicine, Palo Alto, CA 94305, USA; myazdani@stanford.edu (M.Y.); giulia3@stanford.edu (G.B.); 2Laboratorio Cellule Staminali post-natali e Terapie Cellulari, IRCCS G. Gaslini, Via G. Gaslini 5, 16147 Genova, Italy

**Keywords:** γδ T cell, gamma delta T cell, expansion, immunotherapy, allogeneic, third-party, adoptive cell therapy, transduction, bisphosphonate, phosphoantigen

## Abstract

γδ T cells have recently gained considerable attention as an attractive tool for cancer adoptive immunotherapy due to their potent anti-tumor activity and unique role in immunosurveillance. The remarkable success of engineered T cells for the treatment of hematological malignancies has revolutionized the field of adoptive cell immunotherapy. Accordingly, major efforts are underway to translate this exciting technology to the treatment of solid tumors and the development of allogeneic therapies. The unique features of γδ T cells, including their major histocompatibility complex (MHC)-independent anti-cancer activity, tissue tropism, and multivalent response against a broad spectrum of the tumors, render them ideal for designing universal ‘third-party’ cell products, with the potential to overcome the challenges of allogeneic cell therapy. In this review, we describe the crucial role of γδ T cells in anti-tumor immunosurveillance and we summarize the different approaches used for the ex vivo and in vivo expansion of γδ T cells suitable for the development of novel strategies for cancer therapy. We further discuss the different transduction strategies aiming at redirecting or improving the function of γδ T cells, as well as, the considerations for the clinical applications.

## 1. Introduction

γδ T cells are the prototype of ‘unconventional’ T cells, containing a T cell receptor (TCR) composed of γ and δ chains with diverse structural and functional heterogeneity. With both innate- and adaptive-like properties, γδ T cells bridge innate and adaptive immunity [1] and participate in various immune responses to overcome a myriad of challenges. The innate like properties of γδ T cells mirror those of natural killer (NK) cells, expressing the NK receptor NKG2D, and showing cytotoxicity against stressed and abnormal cells, such as viral infected and tumor cells [2]. Interestingly, in most target-response processes, the highly variable γδ TCR is involved which is a distinct feature of adaptive immunity. Human γδ T cells normally comprise only 1–5% of circulating T lymphocytes but undergo rapid expansion in response to tumor, inflammation, and invading pathogens such as cytomegalovirus (CMV) [3] and malaria [4]. In addition, γδ T cells demonstrate cytotoxic activity via the granzyme-perforin axis and antibody (Ab)-dependent cellular cytotoxicity (ADCC) via FcγRIII (CD16) expression [5]. γδ T cells release cytokines such as TNF-α, IFN-γ, and IL-17 [6], prime T cells at the tumor site by playing the role of antigen presenting cells (APC) [7], as well as interacting with B cells in promoting immunoglobulin (Ig) class switching. Several studies have shown that γδ T cells can robustly kill a wide range of tumor cells from both solid and hematopoietic malignancies [8,9] and that their infiltration into the tumors is the most positive prognostic marker in many cancers [10].

γδ T cells are known to recognize stress induced molecules, typically expressed by malignant cells [11]. Ligand recognition by the γδ TCR often requires expression of accessory costimulatory stress molecules on both T lymphocytes and target cells which, therefore, provides a mechanism preventing harmful self-reactivity [12,13]. Unlike conventional αβ T cells, γδ T cells recognize their target antigens irrespective of major histocompatibility complex (MHC) haplotype, and mediate anti-tumor response without causing graft versus host disease (GvHD) [14]. These combined characteristics render γδ T cells ideal candidates for cancer therapy with exciting possible application as a third-party product [15]. Emergence of synthetic biology and novel engineering approaches has provided ample opportunity to genetically modify and manipulate immune cells. These strategies can be employed to optimize the unique anti-tumor function of γδ T cells for future cancer therapies.

### 1.1. Human Adult γδ T Cells Subsets

Human γδ T cells are divided into two main subsets based on their TCR usage of the Vδ1 chain or Vδ2 chain. Most of γδ T cells in human blood use the Vδ2 chain [16] which is typically paired with the Vγ9 chain [17]. Once activated, Vδ2 cells are a source of pro-inflammatory cytokines such as IFN-γ and TNF-α and typically recognize pyrophosphate antigens produced by bacteria [16]. Vγ9Vδ2 T cells account for 50–95% of the peripheral γδ T cells within the 1–5% γδ T cells in circulation. The Vγ9Vδ2 T cells are specially adapted for tumor immunity through potent and broad tumor cytotoxicity, MHC-independency, relative resilience to the suppressive role of programmed cell death-1 (PD-1), low IL-17A production, and activation of NK cells cytotoxicity [18]. Vδ2 γδ T cells have a semi-invariant TCR and mainly behave in an innate-like manner. In adult humans, the γ repertoire of Vγ9Vδ2 T cells is public, whereas the δ repertoire is private [19,20,21].

Vδ1 γδ T cells represent the predominant T cell subset in solid tissues and the second most frequent subset in peripheral blood (1–3% of lymphocytes) next to Vγ9Vδ2 T cells. Vδ1 T cells mainly reside in mucosal epithelial tissues, comprising approximately 40% of all intra-epithelial lymphocytes in the large intestine, and display potent anti-tumor activity. The Vδ1 TCR repertoire in adult humans is often private, primarily consisting of a few clones, sharing adaptive immunity features [22,23,24]. Vδ1 cells via their NKG2D receptors recognize MHC class I related polymorphic molecules such as MICA/MICB [16] and UL16-binding protein (ULBP) which are induced upon stress, damage, or transformation of cells. By engaging NKG2D receptor, MICA/B and ULBP act as a ‘kill me’ signal for the cytotoxic T cells [25] (Figure 1).

Most studies on human γδ T cells have focused on the Vδ2 and Vδ1 subsets, although there are other subsets of γδ T cells with different distribution in various tissues. Of these, the Vδ3 cells represent a minor subset found in blood (≈0.2% of lymphocytes), liver, and gut, that is often described as Vδ1^−^Vδ2^−^. Vδ3 T cells expand in presence of CMV and have been shown to induce dendritic cells (DC) and B cell maturation [26]. Other γδ T cells subsets expressing different Vδ chains such as Vδ4, Vδ5, Vδ6, Vδ7, and Vδ8 [27] have been found which pair with different Vγ chains. Vγ2, Vγ3, Vγ4, Vγ5, and Vγ8 located within the g locus on chromosome 7 in humans, are occasionally used for γ gene rearrangement [28]. These subsets have been found in patients with various infections, but they are rare and have not been sufficiently studied mainly due to the lack of effective expansion techniques and specific antibodies [29]. The current advances on single cell sequencing technology may provide an opportunity to identify and explore these rare subsets within different tissues.

### 1.2. The Hybrid αβ/γδ T Cells

The immunological dogma is that conventional αβ and γδ T cells originate from common precursors in the thymus, then develop into cells expressing either αβ or γδ TCR and occupy a specific and highly conserved niche within the immune system. The mechanisms underlying the divergence and lineage fate are not completely defined. Expression of aberrant TCR chains has been demonstrated among αβ and γδ T cell populations in mice [30,31,32]. For example, in-frame TCR δ rearrangements have been identified in αβ T cells [33], and functional TCR β rearrangements have been detected in γδ T cells [34]. Such aberrant TCR chain expression in mice may be explained by the fact that (i) the murine TCR α/δ locus co-expresses α and δ TCR, thus implying productive rearrangement of TCR α and TCR δ on different alleles, and (ii) the TCR γ locus is repeatedly duplicated which may result in multiple rearrangements. Pellicci et al. characterized a population of human T cells, named δ/αβ T cells, expressing TCRs comprised of a TCR-δ variable gene (Vδ1) fused to joining α and constant α domains, paired with an array of TCR-β chains. They demonstrated that these cells, which represent ≈50% of all Vδ1^+^ human T cells, can recognize peptide- and lipid-based antigens presented by human leukocyte antigen (HLA) and CD1d, respectively [35].

Intriguingly, Ziegler et al. documented a population of Vδ1^+^CD4^+^ γδ T cells expressing stem cell and progenitor markers, i.e., CD34 and CD38, that are able to develop into functional αβ T cells. The route taken by this process involves the re-organization of the Vδ1^+^ γδ TCR into the αβ TCR, as a consequence of TCR-γ chain downregulation and the expression of surface Vδ1^+^Vβ^+^ TCR components. The authors monitored TCR changes in Vδ1^+^CD4^+^ clones and observed that, under inflammatory stimuli, these cells downregulated the TCR-γ and TCR-δ chains and simultaneously rearranged TRBV and TRAV segments. Such trans-differentiation process was readily detectable in vivo in inflamed tissue providing a conceptual framework for extrathymic T cell development. Thus, the αβ T cells reconstituted with this mechanism may be unable to induce GvHD, since they acquire a “self-education” [36].

Interestingly, Bertaina et al. reported that the key signature protein of proteotype γδ T cells in transplanted patients treated with ZOL [37], is Bloom (BLM), which is important in development, maintenance, and function of αβ T lymphocytes [38]. Mutations in *BLM* are responsible for Bloom Syndrome, a disorder characterized by immunodeficiency and propensity to develop cancer. The essential role of BLM in early αβ T cell differentiation was evidenced by the impairment of T cell differentiation, proliferation, and response to antigens in BLM-deficient mice. Thus, in addition to the fact that ZOL increased the Vδ1 percentage and induced BLM in γδ T cells [37], ZOL may induce a ‘reservoir’ of αβ T cell progenitors for the development of αβ T cells in vivo.

Very recently, Edwards et al. identified a discrete population of T cells that coexpressed αβ and γδ TCRs. These hybrid αβ-γδ T cells were transcriptomically distinct from conventional γδ T cells, poised to migrate to sites of inflammation, and were responsive to MHC class I/II-restricted peptide antigens or to stimulation with IL-1β and IL-23. In line with these findings, hybrid αβ-γδ T cells protected against infection with *Staphylococcus aureus* and, by recruiting encephalitogenic Th17 cells, triggered autoimmune pathology in the central nervous system [39]. The hybrid αβ/γδ T cells are a newly discovered population that may illuminate new immunological scenarios and novel therapeutic perspectives.

### 1.3. γδ T Cells: An Appealing Source for Adoptive Cell Immunotherapy

γδ T cells are attractive candidates for adoptive cell immunotherapy due to their unique biology. The following features pinpoint the favorable characteristics of γδ T cells over αβ T cells for cancer treatment.

First, γδ T cell tumor recognition and killing is not dependent on the expression of a single antigen. In contrast, they recognize a broad spectrum of antigens on various cancer cells through their diverse innate cytotoxicity receptors expressed on their cell membrane [40]. This broad response reduces the chances of tumor immune escape by single antigen loss. In addition, this property provides opportunity for designing immunotherapies for tumors lacking well-defined neo-antigens and without the need of further genetic engineering.

Second, γδ T cells recognize their target cells in an MHC-independent manner leading to low or absent risk for alloreactivity and GvHD, thus allowing the development of universal third-party allogeneic cell products for several malignancies.

Third, γδ T cells home in a wide variety of tissues wherein they can rapidly respond to the target and release effector cytokines. This natural tissue tropism of γδ T cells, especially of the Vδ1 subset, provides migratory advantage over αβ T cells and higher ability to infiltrate and function in tumors hypoxic environments [41].

Furthermore, growing evidence indicates that γδ T cells interact with APCs and other immune cells, while also playing the role of APCs by priming the antigens for αβ T cells thereby enabling the orchestration of a cascade of immune responses against tumors [42].

These features make unmodified γδ T cells an attractive source for adoptive cell immunotherapy. However, genetic engineering strategies may also be applied to enhance their cytotoxicity and redirect them toward specific targets. For example, using γδ T cells, either as a vehicle for chimeric antigen receptors (CARs) or αβ T cell-derived TCRs [43], may provide exciting results by combining tissue resident property and innate-like recognition of γδ T cells with antigen-specific activation and engagement of multiple costimulatory signals. To date, the major obstacle to the broad application of γδ T cells for adoptive cell immunotherapy remains effective strategies of in vivo or ex vivo expansion [44,45].

## 2. Expansion Strategies

The broad application of γδ T cells for adoptive cell immunotherapy has been hindered by their low physiological frequency in the periphery, and the difficulty of ex vivo expansion. Considerable efforts are currently devoted to developing suitable methods for obtaining clinical numbers of γδ T cells [45]. The expansion strategy of γδ T cells can be bimodal: ex vivo and in vivo. In the first, γδ T lymphocytes are isolated from peripheral blood mononuclear cells (PBMCs) and stimulated ex vivo using synthetic phosphoantigen (pAg) or bisphosphonates (BP) such as zoledronic acid [46].

Ex vivo expansion of γδ T cells has been clinically applied and has shown promising results [41]. The second approach involves stimulation and expansion of γδ T cells in vivo by systemic administration of pAg or nitrogenous-BP (N-BP). These two approaches will be explained in detail in the following section (Table 1 and Table 2).

Although the main source for isolation of γδ T cells is PBMCs, these cells have also been isolated from alternative sources, such as umbilical cord blood (UCB) [50].

### 2.1. Ex Vivo Expansion of Vδ2 γδ T Cells

Using pAg is the most established method for expanding Vδ2 γδ T cells in vitro and in vivo [17]. The most potent pAg activators for Vδ2 T cells are isopentenyl pyrophosphate (IPP) and (E)-4-hydroxy-3-methyl-but-2-enyl pyrophosphate (HMBPP), the latter being a natural intermediate of the non-mevalonate pathway of IPP biosynthesis [78,79,80]. Small changes in the structure of these pAg, either on the phosphate or the isoprene moieties, profoundly affect the γδ T cell stimulatory capacity. Several analogs of these compounds have been synthesized with an intermediate stimulatory capacity between that of IPP and of HMBPP [81]. The recognition of tumors by γδ T cells is ascribed to the abnormally elevated production of IPP by tumor cells, as result of changes in the regulation of their isoprenoid metabolic pathway [82]. These findings support the concept that γδ T cells may cross-react to phosphorylated metabolites accumulating inside tumor cells and to metabolites released by bacterial cells in the microenvironment. Importantly, when bacteria infect target cells, they induce alteration of the host isoprenoid pathway by subverting several regulatory mechanisms [83,84]. These alterations lead to a transient and acute accumulation of IPP, which is then responsible for the activation of γδ T cells. Therefore, summarizing, γδ T cells may recognize (i) tumor cells that accumulate IPP; (ii) bacterial metabolites such as HMBPP, and (iii) cells accumulating IPP following infection with bacteria not producing HMBPP (Figure 2).

BPs were originally used in the treatment of osteoporosis and other bone disorders [85]; however, they have shown to prompt several clinical indications including cytotoxic activity as a monotherapy [64] and activation of γδ T cells [86]. Zoledronic acid (Zoledronate, Zometa, ZOL) is the most potent of the clinically used N-BP [87]. ZOL has direct effects on tumor cells in vitro through induction of apoptosis and reduction of viability and proliferation of several cancer cell lines [88]. The various natural and synthetic pAg compounds are shown in Figure 2.

Though the molecular recognition events are not yet fully explained, it seems that a hydrophobic alkyl moiety linked to a polar group containing one or more phosphates in these small chemical compounds, is sensed by γδ T cells consequently activating γδ T cells in a TCR-dependent manner [89]. It has been reported that BPs block the isoprenoid biosynthesis pathway by inhibiting farnesyl pyrophosphate synthase (FPPS) enzyme and cause the accumulation of prenyl pyrophosphate metabolites, such as IPP and geranyl pyrophosphate (GPP) within cells [90]. IPP and GPP bind to the cytoplasmic tail of a B7-family related molecule of the Ig-superfamily, called butyrophilin 3A1 (BTN3A1) [91]. This binding stirs a change in BTN3A1 that is detected by Vδ2 TCR at the cell surface, leading to cell proliferation and induction of effector functions [92]. Studies have shown that all the three isoforms of BTN3A molecule, BTN3A1, BTN3A2, and BTN3A3, can stimulate Vγ9Vδ2 T cells [93], and activate them through mechanisms involving multimerization of BTN3A molecules [94]. Other BTN molecules were discovered to play a role in Vδ2 T cell response to pAgs. In a recent study, butyrophilin 2A1 (BTN2A1) was identified as a key ligand binding TCR Vγ9 chain and acting together with BTN3A1 to initiate responses to pAgs. Detection of this heteromeric butyrophilin complexes by Vδ2 T cells represents a distinct class of immune recognition [95].

IPP metabolites can be converted into an ATP analog (ApppI), which could be presented at the cell surface and recognized by γδ TCR [96]; however the exact molecular mechanism is unknown. The proposed mechanisms of γδ T cells activation by pAgs and cellular stress are illustrated in Figure 1.

Vγ9Vδ2 T cells can be simply stimulated and selectively expanded to high cell numbers (up to 800-fold) using pAg or BP plus the appropriate cytokine [61,97]. In clinical studies, the synthetic pAg BrHPP (3 μM) [77] and 2M3B1-PP (100 μM) have been used for ex vivo expansion of Vγ9Vδ2 T cells before adoptive transfer [76]. For expanding Vγ9Vδ2 T cells, ZOL have been used ranging from 1 to 5 μM [98]. A different protocol was tested in a recent study, wherein a much higher concentrations of ZOL (100 μM) was used to pulse PBMC for only a short time (4 h). This protocol resulted in greater expansion compared to those using continuous low concentrations of ZOL (10 μM) [99]. Some of the in vitro studies using pAgs and BPs for γδ T cells expansion are listed in Table 1.

### 2.2. Ex Vivo Expansion of Vδ1 γδ T Cells

To harness polyclonal activity of the γδ T cells, the expansion of a heterogenous population with a broad range of γδ TCRs is highly advantageous. Lorenzo et al. expanded FACS-sorted Vδ1 T cells using a cytokine cocktail and OKT-3 mAb, and demonstrated their robust cytotoxicity against acute myeloid leukemia (AML) cells without inducing antigen loss. Using high-throughput sequencing they confirmed a polyclonal TCR repertoire [40]. Though Vδ1 expansion is not simply achievable via a pharmacological activation comparable to the pAg-induced activation of Vγ9Vδ2 T cells, it has been shown that plant-derived mitogens such as concanavalin-A (Con-A) can expand Vδ1 T cells [42,100]. Siegers et al. achieved a high yield of Vδ1 T cells using Con-A and recombinant human IL-2 and IL-4 (10 ng/mL) on freshly isolated PBMCs, although this result was attributed to the period of Con-A treatment which led to the apoptosis of Vδ2 cells [101].

In addition, a two-step protocol for the expansion of Vδ1 T cells was established by Almeida et al. They used magnetically isolated γδ T cells as starting material, and expanded them for 14 days using anti-CD3 Ab in the presence of IL-1β, IL-4, IL-21, and IFN-γ. Subsequently, cells were transferred into fresh cell culture medium supplemented with IL-15 and IFN-γ, followed by re-stimulation with anti-CD3 Ab and cultured for another week. This expansion protocol resulted in an accumulation of 70% Vδ1 T cells with high expression of NK-receptors and high toxicity against chronic lymphocytic leukemia (CLL) cells [102].

In another study, three different expansion methods including ZOL, Con-A, and CD3/28 antibodies were tested to expand γδ T cells from healthy PBMCs pre-transduction. As expected, ZOL preferentially expanded the Vδ2 subtype (>80% purity by day 13 post-activation). Con-A expanded both Vδ1 and Vδ2 cells; however, the expansion was not as high as ZOL for Vδ2, and most cultured cells remained αβ T cells by day 13. CD3/28 antibodies predominantly expanded αβ T cells as expected [42]. Table 1 shows a summarized list of in vitro studies focused on pAg and BP-based expansion of γδ T cells.

### 2.3. Ex Vivo Expansion Using mAbs

Alternative strategies for expanding or activating γδ T cells that do not respond to pAgs or N-BP, such as the use of monoclonal antibodies (mAb) are in development [41]. A clearer understanding of the Vδ2 TCR signaling and its interaction with BTN3A has led to the development of activating mAbs that potentially remove the need for direct TCR stimulation [103]. Harly et al. demonstrated that 20.1, a weak agonist Ab specific for CD277 (a member of BTN3 subfamily), mimics pAg-induced Vγ9Vδ2 T cell activation [104]. These antibodies may simulate a conformational change in the CD277 molecule that is recognized by Vδ2 TCR [103].

One of the first studies using mAb for expansion of γδ T cells, was performed by Lopez et al. They developed an expansion strategy based on anti-CD2-mAb which generated IL-21-dependant signals. This approach led to the expansion of a large population of viable and functional γδ T cells as well as protecting the γδ T cells from mitogen-induced apoptosis (ADCC). The γδ T cells expanded by this approach, retained their anti-tumor activity against a broad range of hematologic and solid primary tumors and cell lines [105]. To date, the use of anti-γδ TCR mAbs for expansion of γδ T cells has not been successful in vitro. ImCheck Therapeutics is focused on therapeutic Abs that target different members of butyrophilins (BTN/BTNL) family for γδ T cells-based cancer treatment. The activating Abs have been designed to bind certain BTN/BTNL molecules, thereby activating Vγ9Vδ2 T cells for cancer immunotherapy. The antagonist Abs targeting BTN/BTNL molecules are underway for treating auto-immune disease. Currently, there are several companies focusing on generating mAbs for the selective expansion of γδ T cells subset for adoptive cell immunotherapy, but no phase I clinical trials have been opened with these products.

### 2.4. γδ T Cell Modulation with Different Substances

Several substances have been used to modulate γδ T cell differentiation and to augment their anti-tumor reactivity including interleukins, TGF-β, vitamin C, and Abs. PAgs commonly used for the γδ T cells expansion, were found to also skew the profile of Vγ9Vδ2 T cells towards a Th1 like profile, with a characteristic production of IFN-γ and TNF-α [106]. Such Th1-like phenotype can be further shaped by additional factors to serve the needs of immunotherapy. The modulatory effects of different substances on γδ T cells profile are depicted in Figure 3.

#### 2.4.1. Interleukins

IL-2 cytokine has been most frequently used for supporting the expansion and survival of Vγ9Vδ2 T cells [107]. Its working concentration ranged from 100 to 300 IU when combined with pAgs and from 100 to 1000 IU when used with ZOL; and it requires repetitive replenishment in the culture media every 2–3 days [46,98].

Growing evidence indicates that IL-15 is more efficient at expanding γδ T cells with an effector phenotype compared to IL-2. Moreover, effector memory Vγ9Vδ2 T cells derived from renal cell carcinoma tumors were efficiently expanded using BrHPP combined with IL-15, but not IL-2 [108]. In addition, IL-15 induced higher toxicity in BrHPP activated Vγ9Vδ2 T cells against different adherent tumor cells compared to IL-2. This was further supported by the IL-15-mediated overexpression of surface CD56 [109] which is believed to be a marker of γδ T cells cytotoxicity [110]. Similarly, IL-15 in combination with IL-2, boosted the proliferation as well as the in vitro anti-tumor activity of ZOL-expanded Vγ9Vδ2 T cells [111]. Taken together, these studies encourage the use of IL-15 instead of, or in combination with, IL-2 for the expansion of Vγ9Vδ2 T cells, especially with a boosted effector potential, which may enhance the therapeutic efficacy of the Vγ9Vδ2 T cell.

IL-21, a cytokine previously known to enhance NK and CD8^+^ T cell cytotoxicity, was found to significantly promote the proliferation of pAg-induced Vγ9Vδ2 T cells in a dose-dependent manner. IL-21 was shown to enhance proinflammatory response and anti-tumor cytolytic activity of γδ T cells [112]. It also promotes γδ T cells mediated B cell maturation [48]. Vγ9Vδ2 T cells from AML patients exhibited lower expression of IL-21R, which affected their response to IL-21. This was confirmed by the need for a higher dose of IL-21 for expansion and lower increase in STAT1 phosphorylation compared to Vγ9Vδ2 T cells from healthy volunteers. Remarkably, AML Vγ9Vδ2 T cells displayed significantly higher Tim-3 expression compared to healthy Vγ9Vδ2 T cells which was further intensified by pAgs and IL-2. [113]. Interestingly, Tim-3 blockade could restore the proliferation and the STAT1 phosphorylation in Vγ9Vδ2 T cells in response to IL-21. These data indicate that IL-21 could significantly expand the Vγ9Vδ2 T cells, but its potency was restricted due to simultaneously increasing the expression of checkpoint inhibitor, Tim-3 [57].

The central role of interleukin-7 (IL-7) in maintaining the T cell homeostasis is well established [114]. IL-7 exert a pivotal role in modulating γδ T cells function by (a) controlling homeostasis and tissue distribution in both human and mice [115], (b) supporting the development and survival of cutaneous resident γδ T cell population [116] and (c) upregulating γδ T cell’s BTLA expression with consequent apoptosis induction [117]. Moreover, IL-7 is associated with selective expansion of IL-17-producing γδ T cells [118]. In a recent study, involvement of IL-7 in the enrichment of IL-17-producing γδ T cells (γδ17 T cells) in the lymph nodes of old mice was reported [119]. This IL-7/IL-17 synergy is also required for the γδ T cell response to the viral hepatitis infection in vivo [120]. Studies reported that Vδ1 T cells are highly expanded by IL-7, appointing IL-7 as the best candidate cytokine for Vδ1 T cells expansion [121].

#### 2.4.2. Transforming Growth Factor-β (TGF-β)

TGF-β is a pleiotropic cytokine with important role in variety of immune responses. Studies have shown that TGF-β augments the proliferation and cytotoxicity of purified Vδ2 T cells in vitro. In addition, expression of CD54, CD103, INF-γ, IL-9, and granzyme B were upregulated, while CD56 and CD11a/CD18 were downregulated in Vδ2 T cells. Upregulation of E-cadherin-binding molecule, CD103 (αE/β7 integrin) enhanced the γδ T cells synapse formation with tumor cells, and its blockade diminished the TGF-β-induced cytotoxicity of γδ T cell. Altering the adhesion profile of γδ T cells, suggests the essential role of TGF-β in promoting migratory capacity and tissue homing of γδ T cells [109].

#### 2.4.3. Vitamin C

Vitamin C (L-ascorbic acid) is an important vitamin engaged in many different biological processes and is known for its antioxidant and epigenetic modulatory role. Recently, the effect of vitamin C (VC) and its more stable analog, L-ascorbic acid 2-phosphate (pVC) on proliferation and effector function of ZOL- or synthetic pAgs-expanded γδ T cells was investigated. VC or pVC had a relatively mild effect on proliferation of purified γδ T cells and absent effects on ZOL-induced PBMCs. VC and pVC reduced apoptosis, enhanced cellular expansion and cytokine production (e.g., IFN-γ) during primary stimulation of a 14 days period of γδ T cell culture. The modulatory effect of VC and pVC can be harnessed in order to optimize γδ T cells generation for cellular therapy [122].

#### 2.4.4. Monoclonal Antibodies

Studies have shown that Vγ9Vδ2 T cells are capable of mediating Ab-dependent cellular cytotoxicity (ADCC) via their Fc receptor (FcRIIIA, CD16). This mechanism can be exploited by combining mAb and γδ T cells in cancer therapy. Using tumor associated antigens (TAA)-specific Abs, Vγ9Vδ2 T cells can be directed to the tumor site. A pre-clinical study, focusing on human mammary gland carcinoma, showed that treatment with adoptive transfer of BrHPP + IL-2-expanded Vγ9Vδ2 T cells alone had a slight impact on tumor growth; however, when combined with trastuzumab (anti-Her2 Ab), the outcome was significantly improved [123]. Similarly, dinutuximab, an anti-GD2 mAb, increased the tumor lysis activity of γδ T cells by 30% in a neuroblastoma murine model [124].

In a phase I/II clinical trial of low grade follicular lymphoma, rituximab (anti-CD20 Ab) was administered in combination with in vivo BrHPP-activation of Vγ9Vδ2 T cells, and resulted in improved treatment efficacy [125]. Rituximab and two other mAbs (obinutuzumab and daratumumab) were also tested in B cell malignancies in order to improve γδ T cell therapy. Obinutuzumab was found to be most effective in prompting ADCC by γδ T cells against B cell lymphoma cells [98].

Another exciting Ab-based approach is employing bispecific Ab targeting both TAA and Vγ9, hence bridging Vγ9Vδ2 T cells with the tumor cells [126]. Using a bispecific Ab engaging both HER2/neu and γ9 in combination with adoptive transfer of Vγ9Vδ2 T cells in a pancreatic ductal adenocarcinoma xenograft mouse model, a significant reduction in tumor growth was achieved, while the single arm Vγ9Vδ2 T cell therapy was not effective [127].

### 2.5. In Vivo (Systemic) Expansion of γδ T Cells

γδ T cells have complex biology and tissue specific tropism; therefore, they may not behave in in vitro culture as they would under physiologic conditions. This makes interpreting the results of γδ T cells from in vitro studies challenging [128]. To harness the γδ T cell anti-tumor activity, many researchers have attempted to activate γδ T cells in vivo for the treatment of a broad range of tumors. Bertaina et al. have shown that systemic administration of ZOL, to expand Vδ2 γδ T cells in vivo, resulted in increased cytotoxic anti-tumor ability of Vδ2 cells and better overall outcomes in transplanted pediatric leukemia patients. From a clinical point of view, patients who received repeated infusion of ZOL (three or more) showed better survival when compared to those who received 1–2 ZOL infusions [37]. The authors analyzed the effects of ZOL on γδ T cells in vivo, using classical phenotypical and functional assays, synergistically integrated with innovative proteomic tools of sample preparation, as well as analytical conditions including high-resolution mass spectrometry, statistical and network analysis. These novel proteomic approaches applied to clinical studies and high-resolution mass spectrometry revealed an in vivo evolution of γδ T cell proteotype mediated by ZOL. In particular, proteomic analysis of γδ T cells purified from patients showed upregulation of proteins involved in activation processes and immune response, paralleled by downregulation of proteins involved in proliferation. Such effects were already evident after the first ZOL infusion but were further boosted by the subsequent infusions. These outcomes mirrored the phenotypic changes observed through flow cytometry in both Vδ1 and Vδ2 subsets. ZOL influenced, unexpectedly, the phenotype and function not only of Vδ2 cells, which selectively recognize pAgs, but also of the Vδ1 population [37].

Administration of pAgs or N-BP in combination with IL-2 has been a common strategy for expanding Vδ2 γδ T cells in the body. Systemic administration of BrHPP or N-BP (ZOL or pamidronate), in combination with IL-2 has been tested in more than eight clinical trials so far and results have proven safety of the treatment. A successful expansion and maturation of Vδ2 T cells towards IFN-γ-producing effector phenotype was observed in most patients [41].

Data has also demonstrated that IL-2 plays a critical role in the in vivo expansion of γδ T cells. In a study by Dieli et al. ZOL alone or in combination with low-dose IL-2 was used for in vivo expansion of γδ T cells in metastatic hormone-refractory prostate cancer. The results showed no expansion of γδ T cells in the group treated with ZOL alone, whereas in the group treated with ZOL plus IL-2, 5/9 patients showed an increase in γδ T cells population and improved clinical outcome [66]. Some of the clinical studies focused on activating γδ T cells via systemic infusion of pAgs and BPs or adaptive transfer of pAg- and BP-expanded γδ T cells for cancer therapy are listed in Table 2.

## 3. Toward Engineering γδ T Cells: Transduction Strategies

The emergence of synthetic biology has offered a broader set of tools for cellular engineering and reprogramming immune cells. Many engineering strategies have been developed for αβ T cells which might be also used for modifying γδ T cells.

To date, retrovirus has been widely used to transduce γδ T cells mainly due to availability of packaging cell lines which can be stably transduced to produce high virus titer [129]. High transduction efficiencies are achieved using a moloney murine leukemia virus-based vector, SFG [97], along with the RD114 [130] or GALV envelope vectors. To enter the cell nucleus, most retroviruses require the dissolution of the nuclear envelope which only occurs during mitosis. Therefore, retroviruses application becomes mainly restricted to highly proliferating cells [131]. Nevertheless, this has not limited their application for Vδ2 γδ T cells, since they can be activated prior to transduction. Other subsets of γδ T cells such as Vδ1, can be potentially transduced following Con-A driven expansion, but the results has been variable and less predictable [42].

The use of viruses always runs the risk of insertional mutagenesis [132]. Lentiviruses hence have received more attention because of their safer insertional profile [131]. Lamb et al. compared two different lentiviral vectors for transducing γδ T cells and showed a higher transduction efficiency (65% vs. 42%) with simian immunodeficiency virus (SIV) vector compared to human immunodeficiency virus (HIV)-based vector using the same envelope (VSV G) [62]. Wang et al. established a lentiviral-based protocol for transducing γδ T cells after expansion with IPP and IL-2 [58].

To assure that viral gene integration does not occur at the oncogenic loci, while exploiting the potential advantage of targeted integration at a specific loci (TCRα), other editing technologies such as CRISPR/Cas9 have been designed and tested to improve engineered chimeric antigen receptor (CAR) αβ T cells function [133,134]. This strategy has not yet been tested in γδ T cells.

The use of non-viral vectors avoids many of the defenses that cells normally employ against viral vectors. Non-viral transduction methods such as the Sleeping Beauty (SB) Transposon system uses transposase enzymes to insert a new transgene randomly into the target cell [135,136]. Although, transduction efficacy using SB technique has been lower than viruses [137], it provides selective advantages for particular hosts. SB uses electroporation for gene transfer, hence does not require a proliferative status. Deniger at al. used SB to transfer a CD19 CAR gene into a polyclonal population of γδ T which were subsequently expanded using CD19^+^ artificial APCs (aAPCs). These CAR γδ T cells expressing different Vδ genes, displayed enhanced killing of CD19^+^ leukemia xenografts in mice compared with untransduced γδ T cells. SB, unlike proliferation-based transduction methods, did not skew the γδ T cell population toward a particular subset [138]. This may be advantageous when aiming to maintain a particular Vγ/Vδ subset such as non-Vδ2 γδ T cells in order to harness their tropism toward epithelial cells for treating epithelial tumors.

While viral or transposon-based techniques provide stable transduction and therefore long-term persistence of engineered T cells, transient CAR expression strategies have been suggested to reduce CAR T cell toxicity. mRNA transduced CAR αβ T cells were used to target mesothelin in patients [139], although repeated infusions of CAR T cells were required to replenish the reservoir of circulating CAR T cells as expression was lost in few days [43]. Similarly, mRNA transfection by electroporation was used to generate γδ T cells expressing NKT cell derived TCRs [63] and, HLA-A2/gp100-specific TCR or CARs for targeting melanoma. Expression peaked at 24 h and returned to baseline by 72 h post transduction [60]. The in vivo persistence of these cells was not studied.

## 4. Preclinical and Clinical Experience: The Lesson Learned

γδ T cells have been expanded ex vivo and studied in pre-clinical cancer models by numerous groups and their cytotoxicity was demonstrated against a variety of cancer cells derived from breast tumor, lung carcinoma, and liver cancer [140]. A study by Liu et al. showed that γδ T cells are capable of recognizing and lysing prostate cancer cells via innate mechanisms independent of MHC [141]. Due to their high frequency in peripheral blood and easy expansion, Vγ9Vδ2 subset of γδ T cells have been predominantly studied in vitro and in xenograft mouse models of a range of tumors. In general, the best treatment outcomes were achieved when Vγ9Vδ2 cells were expanded ex vivo prior to infusion, when γδ T cell infusion was performed at the same time as tumor cell implantation, or at early timepoints such as once tumors were first palpable (<100 mm^3^ volume), and when the repeated administration of pAg or N-BP drugs plus cytokine (typically IL-2) was implemented.

A handful of clinical trials using γδ T cells for cancer treatment have been performed and several are in progress. Immunotherapies based on Vγ9Vδ2 T cells have been examined for hematological malignancies (NCT02656147), head and neck cancer [142], hepatocellular carcinoma (NCT00562666), renal carcinoma [70,77], mammary carcinoma [143], prostate cancer [66], neuroblastoma [144], and lung cancer [145,146] among others. These clinical studies confirmed the clinical benefit of approaches aiming at activating the anti-tumor cytotoxicity of Vγ9Vδ2 T cells. Notably, they revealed that adoptive transfer of γδ T cells is safe and feasible [147]. In most γδ T cell’s clinical trials, objective responses were observed but the rates of complete remissions were low and long-term disease-free survival data were minimal. The details of different approaches used in γδ T cells clinical trials are beyond the scope of this review, but the lessons we learned from them are briefly described. The overall analysis of the clinical studies shows the safety profile of γδ T cell therapies. However, the efficacy was highly variable. Additionally, it is difficult to compare the efficacy since there is immense variation in the utilized strategies including the different protocols used to expand γδ T cells (ex vivo or in vivo) or the delivery method (e.g., ZOL with or without IL-2 or other cytokines, γδ T cells alone, or in combination with stimulating drugs).

The immunotherapies using pAgs or N-BPs and IL-2 have been successful in increasing the number of circulating Vγ9Vδ2 T cells; nevertheless, there is limited evidence of γδ T cells infiltration into the target tissue. A reduction in the response of circulating Vγ9Vδ2 T cells to the pAgs injection have been observed which may be due to activation-induced anergy [148]. This anergy may have occurred after γδ T cell exposure to the inadequate activatory signals and suboptimal pAgs concentration; although, repeated stimulation of Vγ9Vδ2 T cells by pAgs may also cause terminal differentiation and exhaustion which impedes γδ T cell function [149]. The low potency of pAg-based clinical trials may also be attributed to the poor systemic availability of pAg due to their rapid clearance from plasma. Researchers have attempted to devise new delivering strategies to increase the treatment potency. Local administration of pAgs for eliciting a potent tumor immunity via intratumoral, peritumoral, and intranodal injections have already been tested for various types of cancers. Intranodal injection of pAgs near a tumor mass which allows for activation of γδ T cells outside of the immunosuppressive tumor microenvironment, may improve the treatment efficacy [18]. While it has previously been suggested that γδ T cells are less affected by immunosuppressive tumor microenvironment (TME) due to lower PD1 expression compared to αβ T cell [150], a recent study has introduced a new negative checkpoint receptor—T cell Ig and ITIM domain (TIGIT)—expressed by γδ T cells [151]. Moreover, various immunosuppressive mediators in TME such as TGF-β, indolamine dioxygenase, prostaglandins, or potassium can impair γδ T cells function [107,152,153].

To avoid suboptimal activation of γδ T cells by pAgs in vivo, adoptive transfer of ex vivo expanded γδ T cells have been tested in several clinical trials and shown efficacy, though the number of γδ T cells still diminish over time. To maintain the level and function of the transferred cells, co-administration of ZOL and/or IL-2 have been tested and shown superior efficacy compared to γδ T cells alone [76]. In most of the clinical studies, patient autologous peripheral blood-derived γδ T cells were used; however, in few studies, allogeneic γδ T cells were utilized. Wilhelm et al. used the γδ T cells derived from haploidentical donor for the treatment of hematological malignancies. Three out of four patients showed complete remission with no sign of GvHD, and the γδ T cells persisted for 28 days and expanded in vivo following ZOL and IL-2 injection [71]. A more recent study employed allogeneic γδ T cells for the treatment of patients with cholangiocarcinoma and showed promising results [154]. See Table 2 for a list of clinical studies using pAg and BPs as stimulant for γδ T cells.

## 5. Challenges

Despite considerable efforts, expansion of γδ T cell’s diverse clones and achieving clinically appropriate numbers still pose a major obstacle to the broad application of γδ T cells for adoptive cell immunotherapy [44,45]. γδ T cells hold great potential for cancer therapy; however, an average response rate of only 21% in initial phases of clinical trials denotes limited potency of current γδ T cell therapies [73,74,155]. Immunotherapy based on unmodified γδ T cells removes the potential toxicity that is associated with engineered T cell therapies; though disparities in potency hamper both research and commercial development of γδ-based therapeutics. Additional data and more advanced-phase trials are necessary to determine the efficacy of γδ T cell-based immunotherapies.

There is some controversy regarding the potential γδ T cell tumor-promoting activity via inhibiting anti-tumor responses, enhancing tumor angiogenesis, and secretion of IL-17 [155,156,157]. Whilst recent studies in mice have reported the pro-tumor and pro-metastasis role of the murine IL-17 producing γδ T cells, in humans, IL-17^+^ γδ T cells are rare. Studies have shown that γδ TILs play only a trivial role in secreting IL-17 compared to Th17 and CD4^+^ T cells in the TME [158]. A recent study demonstrated that positively isolated γδ T cells through TCR crosslinking or prolonged stimulation with IPP mediated robust suppressive effects on αβ T cells. In contrast, fresh negatively isolated Vδ2^+^ T cells did not exhibit suppressive behavior, even after stimulation with IL-12/IL-18/IL-15 or the sheer contact with BTN3A1-expressing tumor cells. This data indicates that while pharmacologic stimulation of Vδ2^+^ T cells via Vδ2 TCR for activation and expansion induces Vδ2^+^ T cells’ potent killing activity, it simultaneously promotes suppressive γδ T cells function [56]. Better undertesting of the γδ T cells biology may provide new approaches to polarize pro-tumor γδ T cells toward anti-tumor.

## 6. Conclusions and Future Perspective

To date, preclinical and clinical studies using γδ T cells for treating a variety of cancers have shown great promise. Numerous clinical experiences have convincingly shown that γδ T cells provide a safe and effective platform laying the foundation for allogeneic ‘off-the-shelf’ cell therapies for cancer [41]. The current challenges with the γδ T cells therapy include achieving clinically adoptable numbers and moderate clinical efficacy. The conventional methods of expansion such as phosphonates or mitogens are still suboptimal and able to expand only few subsets of γδ T cells. Using alternative methods of artificial feeder cells have led to expansion of a wider range of γδ T cell subtypes with more versatile TCR clones, though it requires extensive work. Taking advantage of the novel engineering techniques for modifying γδ T cells with growth factor receptors in order to improve their proliferation, survival and persistence while harnessing their intrinsic polyclonal function, could be the future of γδ T cells immunotherapies. The limited efficacy may be addressed with the combinational therapy approaches implementing immune checkpoint inhibitors or mAbs [159].

The development of high-throughput TCR screening and engineering approaches enable the molecular and functional analyses of γδ TCR and provides a better understanding of the dynamic interplay between TCR γ and δ chains with other coreceptors in recognizing and responding to the target antigens and ligands. Identification of cancer cell-sensing γδ TCRs from tumor infiltrating γδ T cells using the modern single-cell isolation and analysis techniques and characterizing their ligands may open novel opportunities for future cell-based cancer therapies.

Advancement in the isolation and expansion of γδ T cells, as well as optimization strategies via genetic engineering approaches paves the way to add γδ T cells to the growing armory of cancer immunotherapeutics. It will be exciting to see the effectiveness of different approaches of γδ T cell immunotherapy in clinical trials especially ‘off-the-shelf’ cell products over the coming years.

## Figures and Tables

**Figure 1 cells-09-01305-f001:**
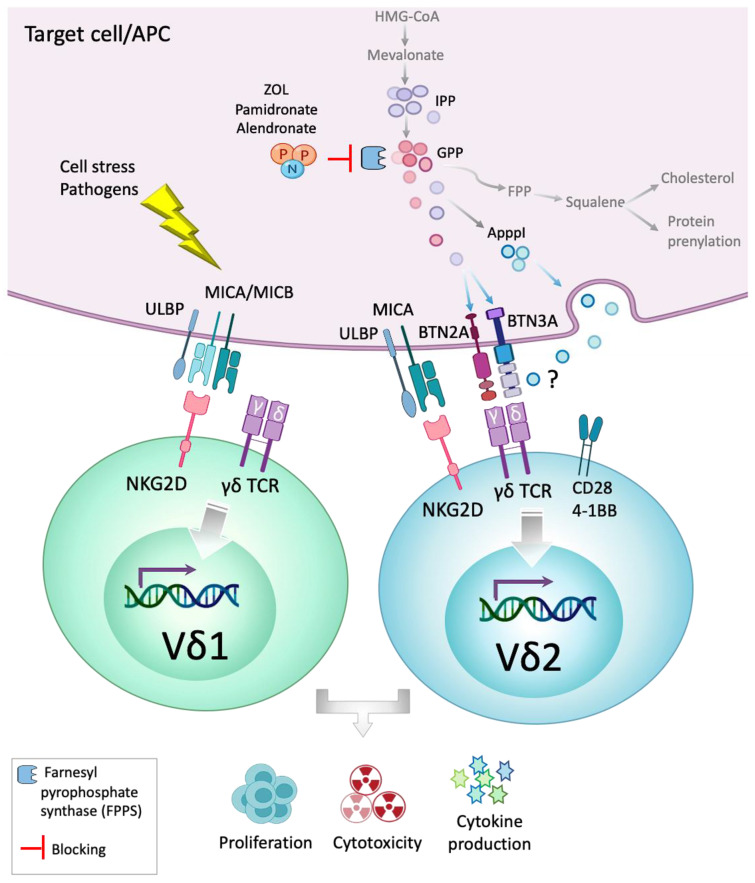
Mechanisms of γδ T cells activation. Bisphosphonates (BP) and nitrogenous-BP (N-BP) block the farnesyl pyrophosphate synthase (FPPS) enzyme in isoprenoid biosynthesis pathway in target cell or antigen presenting cells (APC) which leads to the accumulation of isopentenyl pyrophosphate (IPP) and its metabolites. IPP and geranyl pyrophosphate (GPP) react with cytoplasmic tail of butyrophilin-3 subfamily molecules (e.g., BTN3A1) and stir a change which is detectable by γδ TCR. IPP metabolites can also be converted into an ATP analog (ApppI). ApppI can be presented at the cell surface and be recognized by the γδ TCR; however, the molecular mechanism of this process is not yet clear. Some of the known ligands for Vδ1 and Vδ2 T cells are shown here. Cell stress and bacterial pathogens induce expression of MHC class I chain-related protein A and protein B (MICA/MICB) molecules which react with NKG2D on Vδ1. MICA also binds to NKG2D on Vγ9Vδ2 T cells. Several members of BTN family such as BTN3A1, BTN3A2, and BTN2A can bind to Vδ2 TCR and activate Vδ2 T cells. This activation is often via mechanisms involving multimerization of BTN molecules and exerting a synergistic effect. Moreover, UL16-binding protein (ULBP) which is a ligand for NKG2D receptor, as well as involvement of CD28 and 4-1BB receptors with their ligands provides additional costimulatory signals for γδ T cells.

**Figure 2 cells-09-01305-f002:**
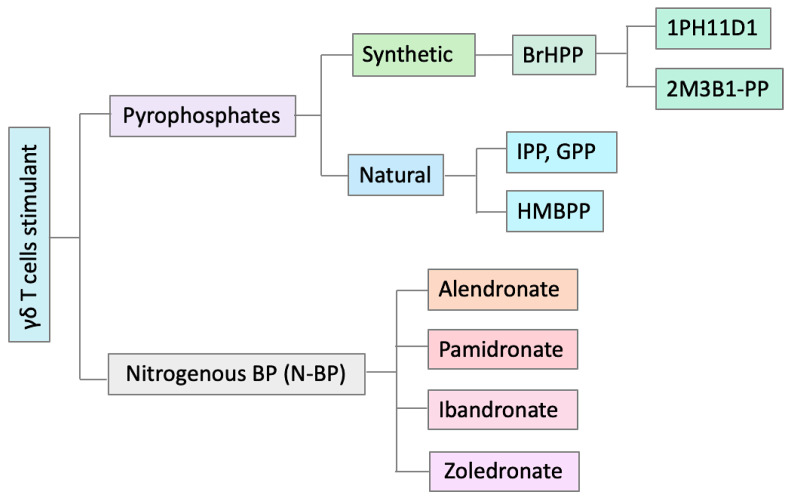
Various phosphoantigen and bisphosphonate compounds. Some of the most important pAgs and BPs mentioned in the text are shown here. BPs are a class of chemical compounds with two PO_3_ (phosphonate) groups that are widely used to treat osteoporosis (the condition of low bone density). Zoledronate is one of the most potent N-BPs that have been widely used for in vitro and in vivo expansion of Vδ2 γδ T cells. (E)-4-hydroxy-3-methyl-but-2-enyl pyrophosphate (HMBPP) is an essential metabolite in most pathogens including mycobacterium tuberculosis and malaria. Isopentenyl pyrophosphate (IPP) and geranyl pyrophosphate (GPP) are the intermediate metabolites in the isoprenoid biosynthesis pathways. Bromohydrin pyrophosphate (BrHPP) is a synthetic alkyl diphosphate pAg. Pyrophosphates such as IPP and GPP are able to directly stimulate Vδ2 γδ T cells, while BPs act indirectly via blocking the FPPS enzyme in isoprenoid biosynthesis pathways which results in IPP and GPP accumulation in the cells (see Figure 1).

**Figure 3 cells-09-01305-f003:**
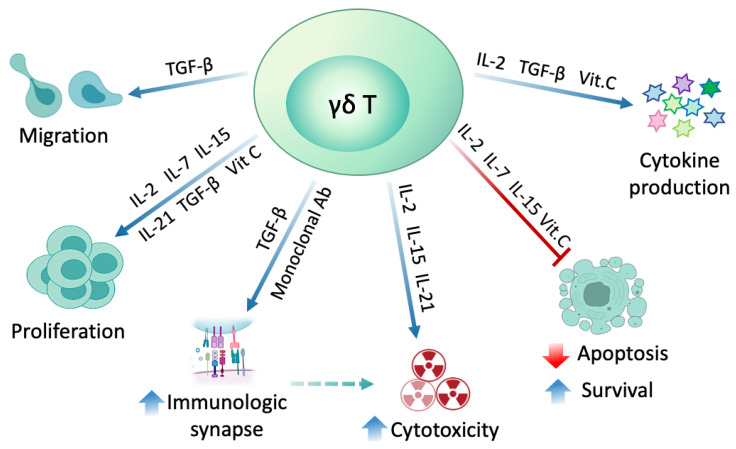
γδ T cells modulation using different substances. Multiple cytokines such as IL-2, IL-7, IL-15, IL-21 and vitamin C skew the profile of γδ T cells toward Th1-like profile meaning increased proliferation, survival, and cytotoxicity. Transforming growth factor-β (TGF-β) enhances Th1-like profile, as well as inducing γδ T cells migration and synapse formation with target cells. Monoclonal antibodies (mAb) and bispecific Abs targeting a tumor associated antigen when used in combination with γδ T cell therapy, can direct γδ T cells to the tumor cells and enhance the formation of cell–cell immunological synapse leading to increased cytotoxicity. Ab, antibody; Vit. C, vitamin C.

**Table 1 cells-09-01305-t001:** In vitro studies using phosphoantigens or bisphosphonates for γδ T cells expansion.

pAg or BP (conc.)	Additional Stimuli	Cytokine (conc.)	Transduction	Subset	Target		Citation	
ZOL (5 uM)		IL-2 (100 IU/mL)	-	-	-	Baker FL. 2020 [47]		
Synthetic HMBPP (0.1–1.0 nM)		IL-2, IL-4, IL-7, IL-15, IL-21, IFNα/β etc.	-	Vγ9Vδ2	-	Vermijlen D. 2007 [48]		
IPP (2 ug/mL)	Irradiated lymphoma cells	IL-12/IL-4 or IL-4/IL-12	-	-	-	Wesch D. 2001 [49]		
ZOL (5 uM)		IL-2 (200 IU/mL)	-	-	Cholangiocarci-noma	Berglund S. 2018 [50] *		
HMBPP (0.1–10 ng/mL)		IL-2, IL-7, Il-15, IL-21	-	-	-	Eberl M. 2002 [51]		
IPP (variable)		IL-2, IL-7, IL-15	-	Vγ9Vδ2	-	Caccamo N. 2005 [52]		
IPP (50 uM)	aAPC, anti-γδ T mAbs	IL-2, IL-21	-	Polyclonal	Neuroblastoma	Fisher J. 2014 [53]		
Pamidronate (10 μg/mL)		IL-2, IL-23, IL-1β, IL-6	-	-	-	Zhang H. 2020 [54]		
ZOL (5 μM), PMA/Ionomycin (750 ng/mL)		-	-	Vδ2, Vδ1	-	Beucke N. 2019 [55]		
ZOL, IPP (20 μg/mL)	Anti-γδ TCR mAb	IL-2, IL-15	-	Vδ2	-	Schilbach K. 2020 [56]		
HMBPP (20 ng/mL)	Feeder cells	IL-2, IL-21	Retroviral	Vγ9Vδ2	-	Wu K. 2019 [57]		
IPP (2–5 ug/mL)		IL-2 (100–1000 U/mL)	Lentiviral	-	-	Wang RN. 2019 [58]		
ZOL (5 uM)		IL-2 (100–200 IU/mL)	Retroviral	-	-	Fisher J. 2019 [59]		
ZOL (40 ug/mL), Con-A (1 mg/mL)		IL-2, IL-4	Retroviral	Vδ2, Vδ1	-	Capsomidis A. 2017 [42]		
ZOL (5 uM), OKT3		IL-2 (1000 IU/mL)	RNA electroporation	-	Melanoma	Harrer DC. 2017 [60]		
ZOL (1 ug/mL)	Irradiated feeder cells	IL-2 (100 IU/mL), IL-15 (10 ng/mL)	Retroviral	-	-	Rischer M. 2004 [61]		
ZOL (1 uM)		IL-2 (50 U/mL)	Lentiviral	-	Glioblastoma	Lamb LS. 2013 [62]		
ZOL (5 μM)		IL-2 (300 IU/mL)	RNA electroporation	Vγ9Vδ2	-	Shimizu K. 2015 [63]		
ZOL (5 uM)	Engineered K562 feeder cells	IL-2 (300 IU/mL)	RNA electroporation	Vγ9Vδ2	-	Xiao L. 2018 [45]		

ZOL, zoledronate; HMBPP, (E)-4-Hydroxy-3-methyl-but-2-enyl pyrophosphate; IPP, Isopentenyl pyrophosphate; Con-A, concanavalin-A; mAb, monoclonal antibody. * Used umbilical cord as source.

**Table 2 cells-09-01305-t002:** Clinical studies using phosphoantigens or bisphosphonates for γδ T cells activation and expansion.

pAg or BP (conc.)	Treatment Strategy	Cytokine	Subset	Target	Citation
ZOL (0.05 mg/kg,1–3 doses)	IV infusion, then in vitro expansion	-	Vδ2, Vδ1	Leukemia	Bertaina A. 2017 [37]
ZOL (4 mg starting dose)	IV infusion + chemotherapy	-	-	Breast cancer	Aft R. 2010 [64]
ZOL (5 uM)	Ex vivo expansion and IP injection	IL-2 (1000 IU/mL)	Vγ9Vδ2	Gastric cancer	Wada I. 2014 [65]
ZOL (4 mg, every 21 days)	IV infusion + Ca and vit. D supplement	IL-2 (0.6 × 106 IU), SQ	-	Prostate cancer	Dieli F. 2007 [66]
ZOL	Ex vivo expansion and adoptive transfer	IL-2 (1000 IU/mL)	-	Non-small cell lung cancer	Nakajima J. 2010 [67], Sakamoto M. 2011 [68]
ZOL (5 uM)	Ex vivo expansion and adoptive transfer	IL-2 (1000 IU/mL)	Vγ9Vδ2	Solid tumors	Noguchi A. 2011 [69]
ZOL (4 mg starting dose)	IV infusion	IL-2 (7 × 10^6^U/m^2^), SQ	Vγ9Vδ2	Renal carcinoma	Lang JM. 2011 [70]
ZOL (4 mg starting dose)	IV infusion post-CD4/CD8 depleted leukapheresis product infusion	IL-2 (1 × 10^6^ U/m^2^), SQ	-	Hematological malignancies	Wilhelm M. 2014 [71]
ZOL (5 uM)	Ex vivo expansion and adoptive transfer	IL-2 (1000 IU/mL)	Vγ9Vδ2	Colorectal cancer	Izumi T. 2013 [72]
ZOL (4 mg starting dose)	IV infusion	IL-2 (2 × 10^6^ IU/m^2^)	-	Renal cell carcinoma, melanoma, acute myeloid leukemia	Kunzmann V. 2012 [73]
Pamidronate (90 mg starting dose)	IV infusion	IL-2 (3 × 10^6^ IU/m^2^)	-	Non-Hodgkin lymphoma or multiple myeloma	Wilhelm M. 2003 [74]
2M3B1-PP (100 uM)	Ex vivo expansion and adoptive transfer	IL-2 (100 IU/mL)	-	Renal carcinoma	Kobayashi H. 2007 [75]
2M3B1-PP (100 uM) + ZOL (4 mg)	Ex vivo expansion and adoptive transfer, + ZOL IV infusion	IL-2 (100 IU/mL), IL-2 (1.4 × 10^6^ IU)		Renal carcinoma	Kobayashi H. 2011 [76]
BrHPP (IPH1101, Phosphostim) (3 uM)	Ex vivo expansion and adoptive transfer	IL-2 (20–60 ng/mL), (2 × 10^6^ IU/m^2^), SQ	Vγ9Vδ2	Metastatic renal cell carcinoma	Bennouna J. 2008 [77]

ZOL, zoledronate; IV, Intravenous; IP, intraperitoneal injection; 2M3B1-PP, 2-methyl-3-butenyl-1-pyrophosphate; BrHPP, Bromohydrin Pyrophosphate; SQ, subcutaneous.

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
