# Peer review of "γδ T Cells: The Ideal Tool for Cancer Immunotherapy"

_cells, 2020, doi:10.3390/cells9051305_

Round 1

Reviewer 1 Report

In this manuscript, the authors described the important roles of gamma/delta T cells in cancer immunotherapy. In general, the manuscript is well-written and would provide sufficient information to the readers in the fields including clinical immunology and oncology. I have few comments as follows.

  1. Several studies indicated that IL-7 regulates gamma/delta T cell homeostatic proliferation. Therefore, the authors should describe the role of IL-7 in gamma/delta T cell modulation.
  2. I suggest that the authors combine section 2.3 (Ex vivo expansion using mAbs) with section 2.4.4 (Monoclonal antibodies).
  3. In line 32, ‘adoptive’ seems to be ‘adaptive’.

Reviewer 2 Report

In this review, an overview is provided of expansion protocols of Vd1 and Vd2 human gd T cells in vitro and in vivo and the different potential transduction strategies aiming at redirecting or improving the function is discussed.

While this is an interesting review, several major and minor issues need to be addressed (detailed below) such as points related to the description of the gd TCR repertoire and the distinction between bisphosphonates and pyrophosphates.

1.

IPP, HMBPP…: bisphosphonates?

Figure 2: needs to be corrected/adapted.

It is confusing and not correct to put HMBPP and other 'non-nitrogenous' molecules under the header of 'bisphosphonates’, that has a rather clinical definition (treatment of bone loss). For example, HMBPP is not used to treat bone loss.

Please correct/adapt.

Related to this: 'Bisphosphonates (BPs) are the chemically stable analogue of pyrophosphate compounds found  in nature'.

This is not correct: for example, zoledronate is not the analogue of HMBPP (zoledronate is not a pyrophosphate). Maybe the authors mean rather BrHPP.

Please verify also Figure 1 regarding this point.

Please adapt also the title of Table 1 regarding this point.

Furthermore, Fig. 1 suggests that MICA/MICB is the only TCR ligand for Vd1+ cells, which is clearly not the case. Maybe the authors can adapt the figure and/or the legend in order to reflect this. Related to this: Line 68: 'Vδ1 cells recognize MHC class I related polymorphic molecules such as MICA/MICB ...' Please clarify. Recognition by the TCR? In Fig. 1 both the Vd1 TCR and NKG2D are interacting with MICA/MICB.; but this is not explained in the text.

2.

TCR repertoire

Line 61-62: 'Vδ2 γδ T cells  have a semi-invariant TCR, a diverse public repertoire, and mainly behave in an innate-like manner'

There are no references here. Furthermore, the delta repertoire of adult Vg9Vd2 T cells is private, not public. Thus as the phrase is written now, it is not correct. The gamma repertoire of adult Vg9Vd2 T cells is public.

Note that the situation in fetal Vg9Vd2 T cells is completely different: here the delta repertoire is public (but the authors probably refer rather to adult V9Vd2 T cells).

Please refer to Ravens 2019 NI, Davey 2018 Nat Comm, Papadopoulou 2019 JI.

Line 66:'The Vδ1 TCR repertoire is often private, primarily consisting of a few clones, sharing adaptive immunity features. '

Also here no references are given. Refer to Ravens 2017 NI, Davey 2017 Nat Comm, Tieppo 2019 JEM.

Line 89: 'Other γδ T cells subsets, such as Vδ4, Vδ5, Vδ6, Vδ8 and Vδ9, have been found which pair with different Vγ chains. Vγ2, Vγ3, Vγ4, Vγ5, Vγ8, Vγ9, and Vγ11 are the 90 most frequently used gene fragments in rearrangement of γ chains, respectively'.

This needs to be corrected. Vd8 and Vd9 subsets? Vg11? (Vg11 in human is not functional, eg Zhang et al (Huck) 1994 EJI). Please verify and adapt.

Line 104: 'TCR α/δ locus co-expresses αβ and γδ TCR... '

Please correct. Locus: a and d (not beta and gamma: are on different chromosomes)

Line 107: 'Although in humans there is a different and more sophisticated TCR structure ...'

What do the authors means with 'more sophisticated TCR structure'? Not sure whether that is the case.

  1. Line 211/paragraph on recognition of phosphoantigens: the authors did not include the two recent important papoers regarding the importance of BTN2A1 (Rigau 2020 Science; Karunakaran 2020 Immuity; summarised by Eberl 2020 Immunol Cell Biol)

Related to this: Figure 1 represents 'CD277' and 'BTN3A1' as separate entities. Please correct.

4.

Line 142: 'First, γδ T cell tumor recognition and killing is not dependent on the expression of a single  antigen. In contrast, they recognize a broad spectrum of antigens on various cancer cells through their  diverse innate cytotoxicity receptors expressed on their cell membrane. This polyclonal response  reduces the chances of tumor immune escape by single antigen loss. ...'

This is a confusing paragraph.

Do the authors refer to the TCR, other receptors or both? 'Polyclonal' is usually used in the context of 'TCR' (not other receptors).

  1. Line 182: paragraph on source of gd T cells for therapy: It is confusing to include mouse skin, next to human peripheral and cord blood mononuclear cells, as a soure of gd T cells in a therapeutic context. The cited studies only investigated mouse skin.

Please rephrase or remove.

  1. Line 232: section on ex-vivo expansion of Vd1:

since the authors highlight the 'polyclonal activity' of this subset, I suggest that they include the study by Di Lorenzo et al 2019 (Cancer Immunol Res), where the authors show the high polyclonal TRD repertoire (by high-throughput sequencing) after in vitro expansion of Vd1+ gd T cells.

  1. Line 254: ‘Alternative strategies for expanding or activating γδ T cells that do not respond to pAgs or N- BP, such as the use of monoclonal antibodies (mAb) are in development [31]. A clearer understanding of the Vδ2 TCR signaling and its interaction with BTN3A has led to the development of activating mAbs that potentially remove the need for TCR stimulation [63]. ‘

This is confusing. Do the authors suggest that stimulation via BTN3A does not depends on the Vg9Vd2 TCR? (is in contrast to their own Fig. 1)

  1. Line 387: 'γδ T cells are predicted to have shorter lifespan 387 compared to αβ T cells ...'

Why? What is the basis for this statement? One could rather say the reverse (see for example Xu 2019 EBioMedicine)

More minor points:

Line 476: 'In most of the clinical studies, patient autologous peripheral blood-derived γδ T cells  were used; however, in a study by Wilhelm et al., allogenic γδ T cells derived from haploidentical  donor was utilized.'

Maybe the authors can refer also to the study by Alnaggar et al2019 J Immunother Cancer.

Line 53: ‘1.1 gd T cell subsets’

I suggest to change this title to 'Human gd T cell subsets' as only human gd T cell subsets are discussed.

Line 122: (start of a new paragraph): 'In support of these findings ...': which finding do the authors refer to? This is not clear.

Lines 42- 43: the authors refer to ligand recognition by gd T cells. Here more relevant references could be used, such as the recent review by Vermijlen et al 2018 (Semin Cell Dev Biol) and Willcox & Willcox 2019 NI).

Line 389: 'To date, retrovirus has been widely used to transduce γδ T cells ...' Is this correct? gd T cells? (reference 81: of 1988)

Line 393: 'Due to the lack of machinery to enter the nucleus, retrovirus function is restricted to highly proliferating cells [83= website]. ' Please verify this statement and use rather a scientific article as a reference.

Line 439: 'A handful of clinical trials using γδ T cells for cancer treatment are already in progress' In progress? (some of the publicatinos are from more than 10 years ago)

Line 267:  ‘The  activating Abs have been designed to bind certain BTN/BTNL molecules, thereby activating different subsets of γ9δ2 T cells for cancer immunotherapy. ‘ What do the authors means with 'subsets of Vg9Vd2 T cells'?

Line 301/paragraph on IL-21: please include the papers by Thedrez et al 2009 JI and Vermijlen et al 2007 JI.

Line 517: 'The development of high-throughput T cell screening ...'.The authors probably mean: '...TCR screening...'

Line 124: this should be ref 29 (Bertaina 2017) instead of ref 28?

Please check carefully the manuscript for English and spelling mistakes.

Round 2

Reviewer 2 Report

Figure 1 is still not correct. Aminobisphosphonates likely activate Vg9Vd2 T cells via the inhibition of the enzyme FPPS (downstream of IPP) leading to intracellular accumulation of the 'real' phosphoantigen IPP. They do not act on mevalonate (upstream of IPP). Why are there two different symbols for FPPS in the figure?

BrHPP is a pyrophosphate. Please check this figure carefully, and make a distinction between the ‘direct’ phosphoantigens such as IPP and the indirect Vg9Vd2 T cell activators such as zoledronate that act by inhibiting FPPS. What is the evidence that etidronate (a ‘nonamino’ bisphosphonate) activate Vg9Vd2 T cells? Please delete this if such evidence is not found. Also from Figure 2.

Fig 2: HMBPP is also a pyrophosphate like IPP! (it is even in the name: PP=pyrophosphate)

Please check carefully the references. For example ref 20 is an erratum of ref 21.

It is confusing to use the term CD277 alongside the BTN nomenclature. It suggests that CD277 is a different molecule than BTN3A1. It would be good to have a  consistent use in the review of the BTN nomenclature.

(for example: ‘Harly et al. 292 demonstrated that 20.1, a weak agonist Ab specific for CD277 (a member of BTN3 subfamily), 293 mimics pAg-induced Vγ9Vδ2 T cell activation (96). These antibodies may simulate a conformational 294 change in the BTN3A molecule that is recognized by Vδ2 TCR (95).’)
